# 3-D Sound Image Reproduction Method Based on Spherical Harmonic Expansion for 22.2 Multichannel Audio

**Kenta Iwai** * [ID]**, Hiromu Suzuki and Takanobu Nishiura** *

College of Information Science and Engineering, Ritsumeikan University, Kusatsu 525-8577, Japan;
is0312ps@ed.ritsumei.ac.jp
* Correspondence: iwai18sp@fc.ritsumei.ac.jp (K.I.); nishiura@is.ritsumei.ac.jp (T.N.)

**Abstract:** In this paper, we propose a three-dimensional (3-D) sound image reproduction method based on spherical harmonic (SH) expansion for 22.2 multichannel audio. 22.2 multichannel audio is a 3-D sound field reproduction system that has been developed for ultra-high definition television (UHDTV). This system can reproduce 3-D sound images by simultaneously driving 22 loudspeakers and two sub-woofers. To control the 3-D sound image, vector base amplitude panning (VBAP) is conventionally used. VBAP can control the direction of 3-D sound image by weighting the input signal and emitting it from three loudspeakers. However, VBAP cannot control the distance of the 3-D sound image because it calculates the weight by only considering the image's direction. To solve this problem, we propose a novel 3-D sound image reconstruction method based on SH expansion. The proposed method can control both the direction and distance of the 3-D sound image by controlling the sound directivity on the basis of spherical harmonics (SHs) and mode matching. The directivity of the 3-D sound image is obtained in the SH domain. In addition, the distance of the 3-D sound image is represented by the mode strength. The signal obtained by the proposed method is then emitted from loudspeakers and the 3-D sound image can be reproduced accurately with consideration of not only the direction but also the distance. A number of experimental results show that the proposed method can control both the direction and distance of 3-D sound images.

**Keywords:** 22.2 multichannel audio; spherical harmonic expansion; 3-D sound image reproduction; mode matching

## 1. Introduction

Three-dimensional (3-D) sound field reproduction systems have become increasingly popular as video technology has advanced. The 3-D sound field reproduction systems are classified with a psychoacoustics-based system and physical acoustics-based system. The binaural system and transaural system are traditional psychoacoustics-based systems [1]. These systems represent the sound pressure at a listener's ears by using a head-related transfer function, which represents the reflection and diffraction by a user's head and torso. In other words, the psychoacoustics-based system represents the direction of the sound image. On the other hand, physical acoustics-based systems, such as wave field synthesis [1], are based on the Kirchhoff–Helmholtz integral and reproduce the sound field by using multiple loudspeakers. In [2], the researchers proposed the sound field reproduction system by using dodecahedron loudspeaker array and achieved the reproducing sound field outside of the loudspeaker array. In [3], higher-order Ambisonics (HOA) is used to reproduce a 2-D sound field in the surrounding area by a circular loudspeaker array and cylindrical loudspeaker array. These researches depict the effectiveness of using loudspeaker arrays to achieve accurate sound field reproduction. In this paper, we focus on a 22.2 multichannel audio [4] as multiple loudspeakers.

22.2 multichannel audio [4] is a 3-D sound field reproduction systems, which has been developed for ultra-high definition television (UHDTV). Figures 1 and 2 and Table 1

show the loudspeaker arrangement of 22.2 multichannel audio, the labels and installation intervals of loudspeakers, and the requirement of the loudspeaker arrangement, respectively. This system can be divided into three layers: upper, middle, and lower. It consists of nine loudspeakers in the upper layer, ten in the middle layer, and three in the lower layer along with two sub-woofers called low frequency effects (LFEs). One of the practical uses of the 22.2 multichannel audio is the theater bar for home use (https://www.nhk.or.jp/strl/open2018/tenji/t2_e.html) (accessed date 20 January 2022). The theater bar is the home reproduction system of 22.2 multichannel audio with a line–array loudspeaker. Recently, the theater bar as a consumer product has been studied in Japan (https://www.jas-audio.or.jp/journal_contents/journal202111_post16264) (accessed date 20 January 2022). According to the ITU-R standards [5], 22.2 multichannel audio can achieve the following effects in sound field.

1. The arrival of sound from all directions surrounding a listening position;
2. High quality 3-D sound impression beyond the 5.1 multichannel audio;
3. High accuracy adjustment of the position between sound and video images.

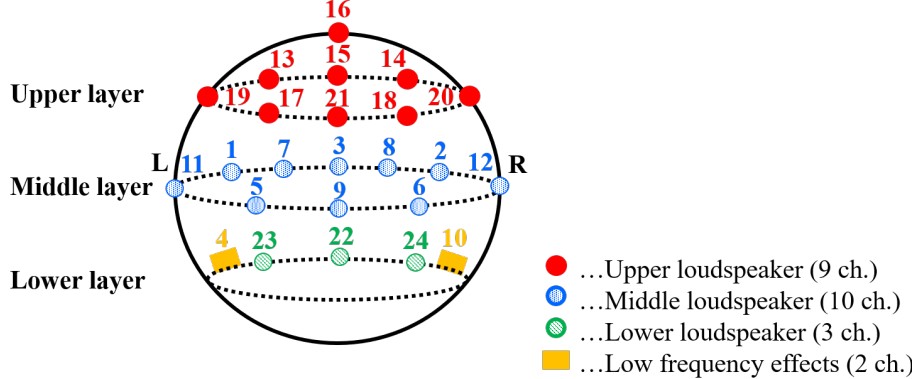

**Figure 1.** Loudspeaker arrangement of 22.2 multichannel audio.

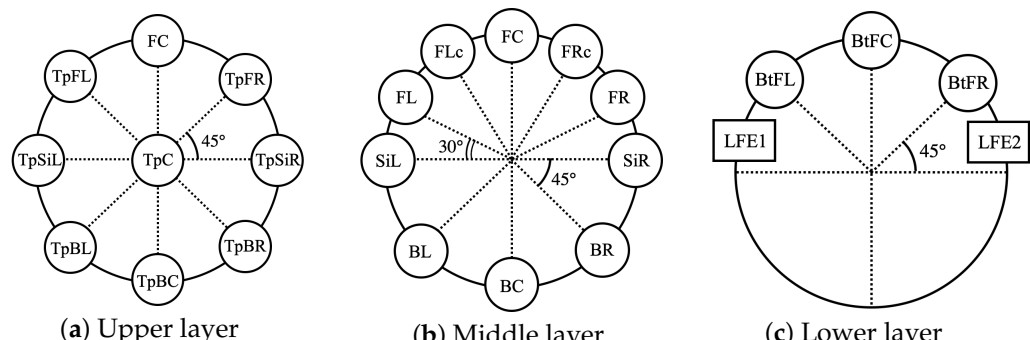

(**a**) Upper layer　　　　　(**b**) Middle layer　　　　　(**c**) Lower layer

**Figure 2.** Labels and installation intervals of loudspeakers.

To achieve these effects, 3-D sound field reproduction systems are generally required to control signals emitted from each loudspeaker. To reproduce the 3-D sound image or field, physical acoustics model-based methods have been studied, for example, in [6–8]. Physical acoustics model-based methods represent the target sound field on the basis of the Kirchhoff–Helmholtz integral equation. In other words, these methods represent the sound field as the physical quantity. In addition, a number of systems adopt the method that represents the arrival direction of the sound [4,9]. The simplest system is the two channels stereophonic system, which is based on the perception of the direction of arrival [10]. Many sound field reproduction systems of this type utilize the panning method to control a 3-D sound image.

**Table 1.** Requirement of loudspeaker arrangement in a 22.2 multichannel audio.

| Layer | Channel No. | Channel Name | Setting Range | |
| | | | Azimuth ($\phi$ [degs.]) | Elevation ($\theta$ [degs.]) |
|---|---|---|---|---|
| Middle | 1 | Front left (FL) | $135 \leq \phi \leq 150$ | $85 \leq \theta \leq 90$ |
| | 2 | Front right (FR) | $30 \leq \phi \leq 45$ | $85 \leq \theta \leq 90$ |
| | 3 | Front center (FC) | 90 | $85 \leq \theta \leq 90$ |
| Lower | 4 | Low frequency effects-1 | $120 \leq \phi \leq 180$ | $105 \leq \theta \leq 120$ |
| Middle | 5 | Back left (BL) | $200 \leq \phi \leq 225$ | $75 \leq \theta \leq 90$ |
| | 6 | Back right (BR) | $315 \leq \phi \leq 340$ | $75 \leq \theta \leq 90$ |
| | 7 | Front left center (FLc) | $112.5 \leq \phi \leq 120$ | $85 \leq \theta \leq 90$ |
| | 8 | Front right center (FRc) | $60 \leq \phi \leq 67.5$ | $85 \leq \theta \leq 90$ |
| | 9 | Back center (BC) | 270 | $75 \leq \theta \leq 90$ |
| Lower | 10 | Low frequency effects-2 | $0 \leq \phi \leq 60$ | $105 \leq \theta \leq 120$ |
| Middle | 11 | Side left (SiL) | 180 | $75 \leq \theta \leq 90$ |
| | 12 | Side right (SiR) | 0 | $75 \leq \theta \leq 90$ |
| Upper | 13 | Top front left (TpFL) | $135 \leq \phi \leq 150$ | $45 \leq \theta \leq 60$ |
| | 14 | Top front right (TpFR) | $30 \leq \phi \leq 45$ | $45 \leq \theta \leq 60$ |
| | 15 | Top front center (TpFC) | 90 | $45 \leq \theta \leq 60$ |
| | 16 | Top center (TpC) | N/A | 0 |
| | 17 | Top back left (TpBL) | $200 \leq \phi \leq 225$ | $45 \leq \theta \leq 60$ |
| | 18 | Top back right (TpBR) | $315 \leq \phi \leq 340$ | $45 \leq \theta \leq 60$ |
| | 19 | Top side left (TpSiL) | 180 | $45 \leq \theta \leq 60$ |
| | 20 | Top side right (TpSiR) | 0 | $45 \leq \theta \leq 60$ |
| | 21 | Top back center (TpBC) | 270 | $45 \leq \theta \leq 60$ |
| Lower | 22 | Bottom front center (BtFC) | 90 | $105 \leq \theta \leq 120$ |
| | 23 | Bottom front left (BtFL) | $135 \leq \phi \leq 150$ | $105 \leq \theta \leq 120$ |
| | 24 | Bottom front right (BtFR) | $30 \leq \phi \leq 45$ | $105 \leq \theta \leq 120$ |

For 22.2 multichannel audio, the conventional panning method, vector base amplitude panning (VBAP) [11,12], can control the direction of the 3-D sound image by vector synthesis. VBAP divides the reproduction space into a triangular area consisting of three loudspeakers and calculates the gains for the respective loudspeakers. However, a VBAP-based system cannot control the distance of 3-D sound images because VBAP considers

only the direction of the 3-D sound image. In [12], the sound intensity is considered to obtain the gain vector for three selected loudspeakers to represent the sound intensity of the sound image with the assumption of locating the real loudspeaker at the position of the sound image. However, both VBAC cannot represent the various directivity pattern because these methods only use a direction vector from the loudspeaker to the sound image. From hereafter, we focus on the original VBAP [11] to simplify the discussion in this paper.

To solve the problem for the original VBAP, we propose a novel 3-D sound image reproduction method based on spherical harmonic (SH) expansion [13]. The proposed method can control both the direction and distance of the 3-D sound image by controlling the sound directivity on the basis of SHs and mode matching [14]. The directivity of the 3-D sound image is obtained in the SH domain and the distance of the 3-D sound image is represented by the mode strength. The signal obtained by the proposed method is then emitted from the loudspeakers and the 3-D sound image can be reproduced accurately. Through two experiments, we evaluate the accuracy of reproduced 3-D sound images between VBAP and the proposed method.

This paper is organized as follows. Section 2 explains the principle of the VBAP as the conventional panning method. In Section 3, the proposed panning method based on SH expansion is explained. A number of experimental results are shown in Section 4. Finally, Section 5 shows the conclusions of this study.

## 2. Conventional 3-D Sound Image Reproduction Based on VBAP

VBAP is a 3-D amplitude panning method based on vector synthesis that localizes any 3-D sound image by using three loudspeakers. By using the position of a sound image and three loudspeakers, the sound image can be reproduced at the desired position. Figure 3 shows an overview of VBAP. In 22.2 multichannel audio, the three loudspeakers, with the exception of the LFEs, are used to reproduce the 3-D sound image by VBAP. The 3-D sound image is reproduced by the following procedure.

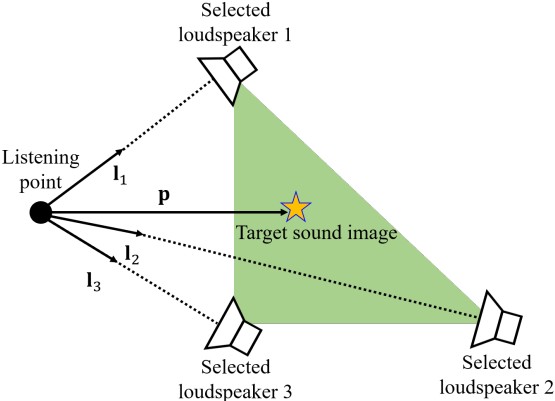

**Figure 3.** Overview of vector base amplitude panning (VBAP).

1.　Obtaining the 3-D sound source position vector **p**.

　　Panning requires the position vector of the 3-D sound image $\mathbf{p} = \begin{bmatrix} p_x & p_y & p_z \end{bmatrix}^{\mathrm{T}}$ to generate signals for reproduction of the sound image. The vector **p** is acquired automatically or manually from media such as video.

2.　Calculation of the gain vector **g**.

　　VBAP calculates the gain vector $\mathbf{g} = \begin{bmatrix} g_1 & g_2 & g_3 \end{bmatrix}^{\mathrm{T}}$ to control the sound image by using **p** and each unit vector $\mathbf{l_1} = \begin{bmatrix} l_{1x} & l_{1y} & l_{1z} \end{bmatrix}^{\mathrm{T}}, \mathbf{l_2} = \begin{bmatrix} l_{2x} & l_{2y} & l_{2z} \end{bmatrix}^{\mathrm{T}}, \mathbf{l_3} = \begin{bmatrix} l_{3x} & l_{3y} & l_{3z} \end{bmatrix}^{\mathrm{T}}$ from the listening point to the three loudspeakers. In VBAP, it is assumed that the 3-D sound image is on the median point among the three loudspeakers. Hence, the position of the 3-D sound image is represented as:

$$\mathbf{p} = \mathbf{L}\mathbf{g}, \tag{1}$$

where $\mathbf{L} = [\mathbf{l_1}\ \mathbf{l_2}\ \mathbf{l_3}]$ is the matrix of unit vectors. From Equation (1), the gain vector $\mathbf{g}$ is obtained by:

$$\mathbf{g} = \mathbf{L}^{-1}\mathbf{p}. \tag{2}$$

3. Normalize the gain vector $\bar{\mathbf{g}}$.
   To prevent excessive sound pressure, the gain vector $\mathbf{g}$ should be normalized by the $L_2$ norm $\|\mathbf{g}\|$ as:

$$\bar{\mathbf{g}} = \frac{\mathbf{g}}{\|\mathbf{g}\|}, \tag{3}$$

where $\bar{\mathbf{g}} = [\bar{g}_1\ \bar{g}_2\ \bar{g}_3]$ is the normalized gain vector.
4. Generation of the input signals $y_i(t)$ of the three loudspeakers.
   The input signals $y_i(t)$ are generated by the object signal $x(t)$ and the calculated gains $\bar{g}_1$, $\bar{g}_2$, and $\bar{g}_3$ as:

$$y_i(t) = \bar{g}_i x(t), \tag{4}$$

where $t$ is the time index and $i \in \{1, 2, 3\}$ is the loudspeaker index.

From these procedures, VBAP can reproduce the 3-D sound image by using the three loudspeakers.

However, VBAP has an issue regarding accurate 3-D sound image reproduction in that it cannot represent the distance of a 3-D sound image. This is because the gain vector $\mathbf{g}$ is calculated by using the radial unit vector. In other words, the gain vector $\mathbf{g}$ only considers the direction of the sound image. To solve this problem, we propose a novel 3-D sound image reproduction method that controls both the directivity and distance of the target sound image on the basis of SH expansion and mode matching.

## 3. Proposed 3-D Sound Image Reproduction Based on Spherical Harmonic Expansion

In this section, we propose a novel 3-D sound reproduction method for 22.2 multi-channel audio based on SH expansion [13]. The proposed method controls the direction and distance of a 3-D sound image by controlling sound directivity on the basis of mode matching [14]. Mode matching generates weighting factors that match the reproduced directivity pattern with the target directivity pattern. In addition, the mode strength used in the mode matching includes the distance of the 3-D sound image. Here, in the spatial sound field, the directivity pattern consists of various types of basic directivity, such as monopole and dipole. On the basis of this fact, SH expansion [15] analyzes the strength of each directivity pattern like a Fourier series expansion. Examples of Fourier series expansion and SH expansion are shown in Figure 4.

The method proposed in [2] has achieved interactive directivity control outside the dodecahedron loudspeaker array using SH expansion, as shown in Figure 5a. The proposed method reproduces the 3-D sound image inside of the 22.2 loudspeakers, as shown in Figure 5b. The proposed method generates a directivity pattern including a peak at the target sound image position. Furthermore, it calculates weighting factors by mode matching in the SH domain. Hence, we can control the amplitude by transforming the obtained weighting factors with consideration of the distance of the 3-D sound image. Here, more than three loudspeakers except for the LFEs are used to reproduce the 3-D sound image. Sections 3.1 and 3.2 explain SH expansion and the proposed method based on it, respectively.

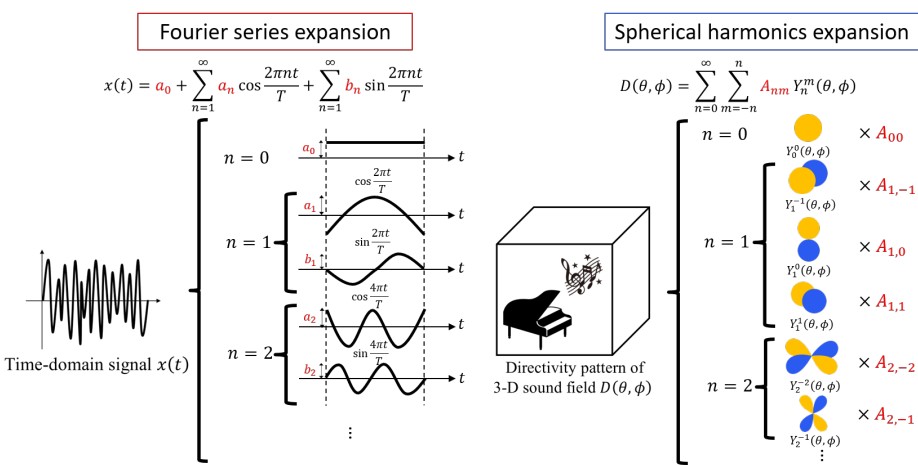

**Figure 4.** Examples of Fourier series expansion and spherical harmonic expansion (SH) expansion.

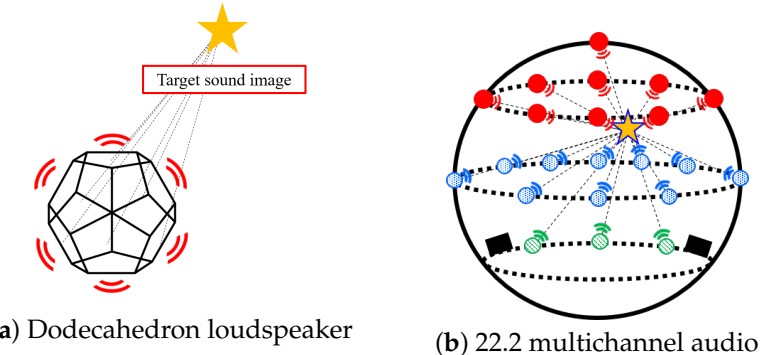

(**a**) Dodecahedron loudspeaker

(**b**) 22.2 multichannel audio

**Figure 5.** Difference between the position of the target sound image on each system.

### 3.1. Spherical Harmonic Expansion

The SH function $Y_n^m(\theta, \phi)$ is a solution of the 3-D wave equation in the spherical coordinate system shown in Figure 6 [15]. It is defined as:

$$Y_n^m(\theta, \phi) = \sqrt{\frac{(2n+1)}{4\pi} \frac{(n-m)!}{(n+m)!}} P_n^m(\cos\theta) e^{jm\phi}, \tag{5}$$

where $n(0 \leq n \leq \infty)$ is the order of the SH function, $m(-n \leq m \leq n)$ is the degree of the SH function, $\theta(0 \leq \theta \leq \pi)$ is the elevation angle, $\phi(0 \leq \phi \leq 2\pi)$ is the azimuth angle, and $P_n^m(\cdot)$ is the Legendre function. Figure 7 shows the shapes of the SH function at the order of $n = 2$ and degree of $m = 2$ in Cartesian coordinates. The color of the SH function means phase $\angle Y_n^m(\theta, \phi)$: yellow and blue show positive and negative values, respectively. This function can be applied to an orthogonal function expansion to determine the arrival direction of a plane wave. Therefore, any directivity pattern $D(\theta, \phi)$ can be expanded using the SH function $Y_n^m(\theta, \phi)$ and coefficient $A_{nm}$ in the SH domain as:

$$D(\theta, \phi) = \sum_{n=0}^{\infty} \sum_{m=-n}^{n} A_{nm} Y_n^m(\theta, \phi). \tag{6}$$

Here, $A_{nm}$ indicates the strength of each basic directivity. In the SH domain, it is possible to analyze what percentage of the desired directivity pattern $D(\theta, \phi)$ includes basic directivity, such as monopole and dipole. From Equation (6), $A_{nm}$ can be calculated as:

$$A_{nm} = \int_0^\pi \int_0^{2\pi} D(\theta, \phi) Y_n^m(\theta, \phi)^* \sin\theta d\theta d\phi, \tag{7}$$

where $*$ is the complex conjugate.

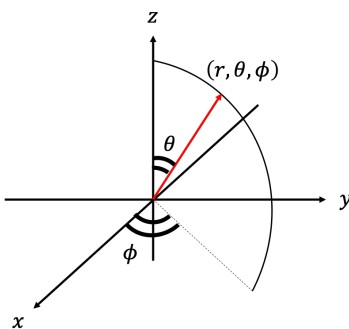

**Figure 6.** Spherical coordinate system.

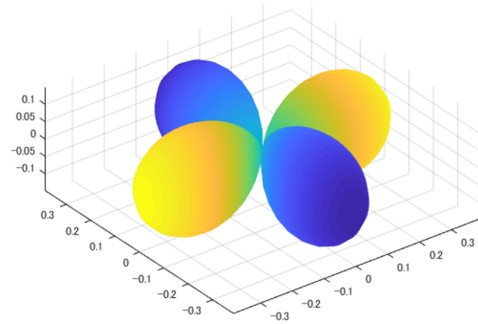

**Figure 7.** Shapes of SH function at order $n = 2$ and degree $m = 2$.

*3.2. Algorithm of Proposed Method*

We explain the procedures of the proposed method to generate an accurate 3-D sound image.

1.  Generation of a target directivity pattern $D^{\text{Tar}}(\theta, \phi)$.

    The proposed method requires a target directivity pattern $D^{\text{Tar}}(\theta, \phi)$ including a peak at the target sound image position. This directivity pattern affects the clarity of the 3-D sound image at the target position. In this paper, a provisional directivity pattern is generated by multiple signal classification (MUSIC) [16]. MUSIC estimates the direction of arrival of a sound source and generates the sharp spatial spectrum to the direction of the sound source. In Step 1, we generate the spatial spectrum $P(\theta, \phi)$ for the target sound image at the position of $(r^{\text{Tar}}, \theta^{\text{Tar}}, \phi^{\text{Tar}})$ by using MUSIC. Then, the generated spatial spectrum $P(\theta, \phi)$ is normalized to the target directivity pattern $D^{\text{Tar}}(\theta, \phi)$ as:

$$D^{\text{Tar}}(\theta, \phi) = \frac{P(\theta, \phi) - P(\theta^{\text{Min}}, \phi^{\text{Min}})}{P(\theta^{\text{Tar}}, \phi^{\text{Tar}}) - P(\theta^{\text{Min}}, \phi^{\text{Min}})}, \tag{8}$$

    where $(\theta^{\text{Min}}, \phi^{\text{Min}})$ is the position with the smallest spatial spectrum. An example of the target directivity pattern $D^{\text{Tar}}(\theta, \phi)$ is shown in Figure 8.

2.  Calculation of the target directivity pattern in the SH domain $A_{nm}^{\text{tar}}$.

    We calculate the target directivity pattern in the SH domain $A_{nm}^{\text{Tar}}$ using Equations (5) and (7) as:

$$A_{nm}^{\text{Tar}} = \int_0^\pi \int_0^{2\pi} D^{\text{Tar}}(\theta, \phi) Y_n^m(\theta, \phi)^* \sin\theta d\theta d\phi. \tag{9}$$

3.  Calculation of the weighting factor $w_i(k)$.

We calculate the weighting factor $w_i(k)$ for $i$th loudspeaker at the position of $(r_i^{\text{LS}}, \theta_i^{\text{LS}}, \phi_i^{\text{LS}})$ to reproduce the target directivity pattern $A_{nm}^{\text{Tar}}$ in the real sound field. The weighting factor $w_i(k)$ is obtained by:

$$w_i(k) = \sum_{n=0}^{\infty} \sum_{m=-n}^{n} w_{nm,i}(k) Y_n^m(\theta_i^{\text{LS}}, \phi_i^{\text{LS}}), \tag{10}$$

where $k$ is the wave number and $i \in \{1, 2, \cdots, 22\}$ is the loudspeaker index. On the basis of the mode matching, the target directivity pattern $A_{nm}^{\text{Tar}}$ has a relationship between the weighting factor $w_{nm,i}(k)$ in the SH domain as:

$$w_{nm,i}(k) = \frac{A_{nm}^{\text{Tar}}}{b_n^S(k, r_i^{\text{LS}})}, \tag{11}$$

$$b_n^S(k, r_i^{\text{LS}}) \triangleq \frac{4\pi}{(r_i^{\text{LS}})^2 k} \frac{j_n(kr^{\text{Tar}})}{j_n{}'(kr_i^{\text{LS}})}, \tag{12}$$

where $j_n(\cdot)$ and $j_n'(\cdot)$ are the spherical Bessel function and its derivative, respectively, and $b_n^S(\cdot)$ is the mode strength [17]. The mode strength $b_n^S(\cdot)$ theoretically represents the radial strength of the directivity. Substituting Equation (11) into Equation (10), the weighting factor $w_i(k)$ can be obtained as follows:

$$w_i(k) = \sum_{n=0}^{\infty} \sum_{m=-n}^{n} \frac{A_{nm}^{\text{Tar}}}{b_n^S(k, r_i^{\text{LS}})} Y_n^m(\theta_i^{\text{LS}}, \phi_i^{\text{LS}}). \tag{13}$$

4.  Generation of the input signals for 22 loudspeakers.
    The weight factor $w_i(k)$ in the spatial domain can be used as the frequency domain filter, that is,

$$W_i(\omega) = w_i\left(\frac{\omega}{c}\right), \tag{14}$$

$$k = \frac{\omega}{c}. \tag{15}$$

where $\omega$ is the angular frequency and $c$ is the speed of sound. Then, the input signal for $i$th loudspeaker $y_i(t)$ is obtained by:

$$y_i(t) = \text{IDTFT}[W_i(\omega)X(\omega)], \tag{16}$$

$$X(\omega) = \text{DTFT}[x(t)], \tag{17}$$

where $x(t)$ is the sound source, IDTFT[$\cdot$] and DTFT[$\cdot$] represent the inverse discrete time Fourier transform and discrete time Fourier transform operators, respectively.

By inputting $y_i(t)$ to each $i$th loudspeaker, the 3-D sound image can be reproduced with consideration of not only the direction but also the distance of the sound image. For the 3-D sound image reproduction, the maximum order of the SH expansion is generally limited as:

$$D(\theta, \phi) = \sum_{n=0}^{N} \sum_{m=-n}^{n} A_{nm} Y_n^m(\theta, \phi), \tag{18}$$

$$N = \lceil k_{\max} R \rceil, \tag{19}$$

where $k_{\max} = \omega_{\max}/c$ is the maximum value of the wave number, $R$ is the radius of the reproduced sound field, and $\lceil \cdot \rceil$ is the ceiling function. Here, $k_{\max}$ represents the largest value of the wave number for the target sound. Hence, the order $n$ in Equation (11) is also limited to $N$.

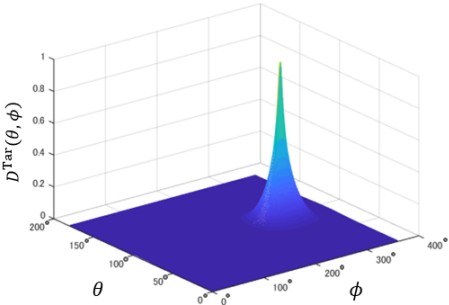

**Figure 8.** Example of directivity pattern $D^{\mathrm{tar}}(\theta, \phi)$ obtained by multiple signal classification (MUSIC).

Regarding the computational complexity, the proposed method requires the calculation of the directivity pattern of the sound image, coefficients of SH expansion, and mode strength for obtaining the filter coefficients. On the other hand, the conventional VBAP requires only vector calculation. Hence, the proposed method has higher computational complexity than that of the conventional method.

## 4. Evaluation Experiment

We conducted a number of experiments to evaluate the effectiveness of the proposed method. In the following experiments, we recorded the outputs of the 22.2 multichannel audio using the signal generated by VBAP and the proposed SH expansion-based method. From hereafter, we use the notation "conventional method" for VBAP and "proposed method" for the proposed SH expansion-based method.

The experimental environment and equipment are shown in Table 2. The loudspeakers were placed at the positions shown in Table 3. Here, the listening point was the center of the surrounded area by the loudspeakers, 1.2 m in height from the floor. In addition, each loudspeaker was 1.9 m away from the listening point.In these experiments, we used band-limited white noise with a duration of 3.0 s as the sound source for the conventional and proposed methods. The frequency band of the band-limited white noise was set to 0–8000 Hz. This is because the signal with high frequency components should be represented with the higher-order spherical harmonic expansion, however, it is difficult to obtain the coefficients for SH expansion. Furthermore, in accordance with the highest frequency of the sound source, the maximum order of the SH expansion $N$ in Equation (18) was set to 5 for all experiments. The positions of each target sound image are shown in Table 4. Here, (a) and (b), (c) and (d), (e) and (f), and (g) and (h) are, respectively, the same direction and different distance of each target sound image. The emitted sound was recorded by the dummy head microphone placed at the position shown in Table 5.

**Table 2.** Experimental environment and equipment.

| | |
|---|---|
| Environment | Experiment room ($T_{60} = 300$ ms) |
| Ambient noise level | 37.0 dBA |
| Sound pressure level | 70.0 dB at the listening point |
| Dummy head | 3Dio, Free Space Pro II |
| Loudspeaker | YAMAHA, VXS5 |
| Loudspeaker (LFE) | YAMAHA, VXS10S |
| Loudspeaker amplifier | YAMAHA, XMV8280 |
| Analog-to-digital converter | RME, Fireface UFX |
| Digital-to-analog converter | RME, M-32 DA |

**Table 3.** Loudspeaker arrangement of 22.2 multichannel audio used in the experiment.

| Layer | Channel No. | Channel Name | Setting Position | |
|---|---|---|---|---|
| | | | Azimuth ($\phi$ [degs.]) | Elevation ($\theta$ [degs.]) |
| Middle | 1 | Front left (FL) | 150 | 90 |
| | 2 | Front right (FR) | 30 | 90 |
| | 3 | Front center (FC) | 90 | 90 |
| Lower | 4 | Low frequency effects-1 | 150 | 118.3 |
| Middle | 5 | Back left (BL) | 210 | 90 |
| | 6 | Back right (BR) | 330 | 90 |
| | 7 | Front left center (FLc) | 120 | 90 |
| | 8 | Front right center (FRc) | 120 | 90 |
| | 9 | Back center (BC) | 270 | 90 |
| Lower | 10 | Low frequency effects-2 | 30 | 118.3 |
| Middle | 11 | Side left (SiL) | 180 | 90 |
| | 12 | Side right (SiR) | 0 | 90 |
| Upper | 13 | Top front left (TpFL) | 150 | 52 |
| | 14 | Top front right (TpFR) | 30 | 52 |
| | 15 | Top front center (TpFC) | 90 | 52 |
| | 16 | Top center (TpC) | - | 0 |
| | 17 | Top back left (TpBL) | 210 | 52 |
| | 18 | Top back right (TpBR) | 330 | 52 |
| | 19 | Top side left (TpSiL) | 180 | 52 |
| | 20 | Top side right (TpSiR) | 0 | 52 |
| | 21 | Top back center (TpBC) | 270 | 52 |
| Lower | 22 | Bottom front center (BtFC) | 90 | 118.3 |
| | 23 | Bottom front left (BtFL) | 150 | 118.3 |
| | 24 | Bottom front right (BtFR) | 30 | 118.3 |

**Table 4.** Position of target sound image $(r^{\text{Tar}}, \theta^{\text{Tar}}, \phi^{\text{Tar}})$.

| | |
|---|---|
| (a) | $(1.0, 45°, 45°)$ |
| (b) | $(1.5, 45°, 45°)$ |
| (c) | $(1.0, 45°, 135°)$ |
| (d) | $(1.5, 45°, 135°)$ |
| (e) | $(1.0, 45°, 225°)$ |
| (f) | $(1.5, 45°, 225°)$ |
| (g) | $(1.0, 45°, 315°)$ |
| (h) | $(1.5, 45°, 315°)$ |

**Table 5.** Position of the dummy head microphone $(r, \theta, \phi)$.

| | |
|---|---|
| A | $(0.5, 90°, 180°)$ |
| B | $(0.5, 90°, 225°)$ |
| C | $(0.5, 90°, 270°)$ |
| D | $(0.5, 90°, 315°)$ |
| E | $(0.5, 0°, 90°)$ |

*4.1. Experiment 1: Evaluation of Sound Image Localization Accuracy*

In this experiment, the sound image localization accuracy was evaluated on the basis of [18]. The evaluation method [18] utilizes the head-related transfer function (HRTF) database and the direction of the sound image $(\hat{\theta}, \hat{\phi})$ can be estimated accurately. However, the distance between the recorded position and sound image cannot be obtained directly by this evaluation method. Hence, we estimated the position of the sound image $(\hat{r}, \hat{\theta}, \hat{\phi})$ by using the image's estimated direction and the recorded sounds at five positions shown in Table 5. The evaluation procedure is shown as follows:

1. Calculation of inter-aural level difference (ILD) and inter-aural phase difference (IPD)
   The ILD and IPD were calculated for each recorded sound at positions A to E shown in Table 5 [19]. Here, the ILD and IPD are known as factors for the sound localization of humans. The ILD and IPD can be calculated by:

$$\text{ILD}_\Omega(\omega) = 20 \log_{10} \left| \frac{C_\Omega(\omega)}{P_\Omega(\omega)} \right|, \tag{20}$$

$$\text{IPD}_\Omega(\omega) = \tan^{-1} \left( \frac{\text{Im}(C_\Omega(\omega))}{\text{Re}(P_\Omega(\omega))} \right), \tag{21}$$

where $C_\Omega(\omega)$ is the cross spectrum between the signals obtained at the left and right ears of the dummy head microphone, $P_\Omega(\omega)$ is the power spectrum of the signal obtained at the left ear of the dummy head microphone, and $\text{Re}(\cdot)$ and $\text{Im}(\cdot)$ represent the real and imaginary parts of the complex value, respectively. $\Omega \in \{A, B, C, D, E\}$ is the index for the recording position.
   In addition, $\text{ILD}_{\text{HRTF}}(\omega, \theta, \phi)$ and $\text{IPD}_{\text{HRTF}}(\omega, \theta, \phi)$ were calculated by using HRTF in the CIPIC database [20]. $\text{ILD}_{\text{HRTF}}(\omega, \theta, \phi)$ and $\text{IPD}_{\text{HRTF}}(\omega, \theta, \phi)$ represent the value for which the sound image perfectly localizes at the desired position.

2. Calculation of the differences of the ILD and IPD between recorded sound and HRTF database.
   The differences of the ILD and IPD between the recorded sound and HRTF database were calculated. If the differences are small, it can be said that the sound image localizes at the direction $(\theta, \phi)$. The difference between the inter-aural time difference (ITD) and IPD are defined as:

$$E_{\Omega,\text{ILD}}(\omega, \theta, \phi) = |\text{ILD}_\Omega(\omega) - \text{ILD}_{\text{HRTF}}(\omega, \theta, \phi)|, \tag{22}$$

$$E_{\Omega,\text{IPD}}(\omega, \theta, \phi) = |\text{IPD}_\Omega(\omega) - \text{IPD}_{\text{HRTF}}(\omega, \theta, \phi)|. \tag{23}$$

Then, $E_{\Omega,\text{ILD}}(\omega,\theta,\phi)$ and $E_{\Omega,\text{IPD}}(\omega,\theta,\phi)$ were combined as the following form.

$$E_{\Omega}(\omega,\theta,\phi) = \beta(\omega)E_{\Omega,\text{IPD}}(\omega,\theta,\phi) + (1-\beta(\omega))E_{\Omega,\text{ILD}}(\omega,\theta,\phi), \qquad (24)$$

$$\beta(\omega) = \begin{cases} 1 & (\omega \le \omega_{\text{L}}) \\ 1 - \frac{\omega-\omega_{\text{L}}}{\omega_{\text{H}}-\omega_{\text{L}}} & (\omega_{\text{L}} < \omega < \omega_{\text{H}}), \\ 0 & (\omega \ge \omega_{\text{H}}) \end{cases} \qquad (25)$$

where $\beta(\omega)$ is the weighting function that controls the ratio between $E_{\Omega,\text{IPD}}(\omega,\theta,\phi)$ and $E_{\Omega,\text{ILD}}(\omega,\theta,\phi)$ in Equation (24). $\beta(\omega)$ has a characteristic shown in Figure 9. The reason for using $\beta(\omega)$ is that the IPD affects the sound localization of humans below 1500 Hz and is dominant above 1500 Hz [21]. Hence, we set $\omega_{\text{L}} = 6283$ rad/s and $\omega_{\text{H}} = 12{,}566$ rad/s with consideration of the crossover. These values are related to 1000 and 2000 Hz in terms of frequency. From hereafter, $E_{\Omega}(\omega,\theta,\phi)$ is called the error function.

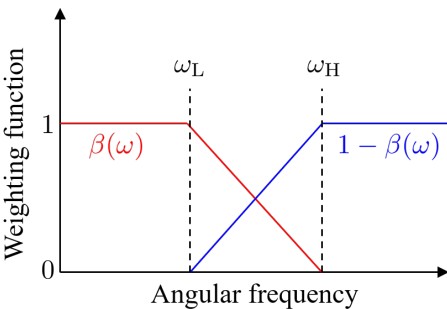

**Figure 9.** Weighting function $\beta(\omega)$ and $1 - \beta(\omega)$.

3.  Estimation of the direction of reconstructed 3-D sound image.
    The direction of the reconstructed 3-D sound image can be estimated by obtaining $(\hat{\theta}_{\Omega}, \hat{\phi}_{\Omega})$ of which the error function $E_{\Omega}(\omega,\theta,\phi)$ is the smallest value. Here, the directions of the sound image were estimated for each recorded sound as:

$$(\hat{\theta}_{\Omega}, \hat{\phi}_{\Omega}) = \underset{\substack{0 \le \theta \le 180° \\ 0 \le \phi \le 360°}}{\arg\min} \left| \int_{0}^{\omega_{\max}} E_{\Omega}(\omega,\theta,\phi)d\omega \right|, \qquad (26)$$

where $\omega_{\max}$ is the highest angular frequency of the recorded sound. In this experiment, the frequency of the sound is up to 8000 Hz and $\omega_{\max} = 50265$ rad/s.
4.  Estimation of the position of the reconstructed 3-D sound image.
    Finally, the position of the reconstructed 3-D sound image was estimated by using the estimated direction of the sound image $(\hat{\theta}_{\Omega}, \hat{\phi}_{\Omega})$. As shown in Figure 10, the position was estimated by drawing a straight line from each recording position to the estimated direction. Then, the center position of the surrounding area by the five lines was treated as the estimated position of the sound image $(\hat{r}, \hat{\theta}, \hat{\phi})$.

From the four procedures, the position of the 3-D sound image was estimated and the accuracy of the sound image localization for the proposed method was evaluated by conducting two trials of Experiment 1.

Moreover, the error between the positions of the reconstructed sound image and that of the target sound was evaluated. The error $D_{\text{Err}}$ is defined as:

$$D_{\text{Err}} = \sqrt{(\Delta X)^2 + (\Delta Y)^2 + (\Delta Z)^2}, \tag{27}$$

$$\Delta X = \hat{r}\sin\hat{\theta}\cos\hat{\phi} - r^{\text{Tar}}\sin\theta^{\text{Tar}}\cos\phi^{\text{Tar}}, \tag{28}$$

$$\Delta Y = \hat{r}\sin\hat{\theta}\sin\hat{\phi} - r^{\text{Tar}}\sin\theta^{\text{Tar}}\sin\phi^{\text{Tar}}, \tag{29}$$

$$\Delta Z = \hat{r}\cos\hat{\theta} - r^{\text{Tar}}\cos\theta^{\text{Tar}}. \tag{30}$$

By Equation (27), the error in terms of the Euclidean distance can be evaluated. The error $D_{\text{Err}}$ was calculated for each trial. Then, the average of the errors was evaluated.

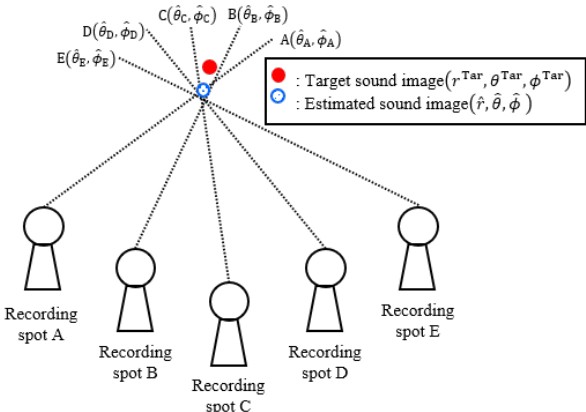

**Figure 10.** Estimation of the position of the reconstructed 3-D sound image $(\hat{r}, \hat{\theta}, \hat{\phi})$.

Figure 11 shows the estimated position of the reconstructed sound image for each condition (a)–(h) on the first trial and Figure 12 shows the error $D_{\text{Err}}$ for each position. In Figure 11, the closer the red and black markers, the higher the accuracy of the sound localization. From Figures 11 and 12, the proposed method can reconstruct the sound image close to the target position. The distance error $D_{\text{Err}}$ of the proposed method is 0.17 m smaller than that of the conventional method on average. These results show the effectiveness of the proposed method, which includes not only the elevation angle $\theta$ and the azimuth angle $\phi$ but also the distance $r$. Here, the error $D_{\text{Err}}$ of the proposed method is about 0.87 m at positions (a), (c), (e), and (g) at which the distance of the target sound image is 1.0 m, although the error $D_{\text{Err}}$ of the conventional method is about 1.02 m. However, the error $D_{\text{Err}}$ of the proposed method is about 1.30 m at positions (b), (d), (f), and (h) at which the distance of the target sound image is 1.5 m, although the error $D_{\text{Err}}$ of the conventional method is about 1.49 m. These results indicate that the sound pressure and phase largely affect the sound localization in the case that the recording position, that is, the listening position, is close to the position of the sound image. Hence, the improvement of sound localization is more significant at positions (a), (c), (e), and (g) than that at positions (b), (d), (f), and (h).

Focusing on the direction of the sound image, the error $D_{\text{Err}}$ of the proposed method is about 0.90 m at positions (a)–(d), although the error $D_{\text{Err}}$ of the conventional method is about 1.13 m. In other words, the proposed method accurately reconstructs the 3-D sound image in front of the listening position. However, the error $D_{\text{Err}}$ of the proposed method is about 1.27 m at positions (e)–(h), although the error $D_{\text{Err}}$ of the conventional method is about 1.39 m. In other words, the accuracy of the reconstruction of the 3-D sound image behind the listening position degrades for both the conventional and proposed methods. This is because the dummy head microphone represents a human auditory system, that is, the accuracy of the sound localization behind the listening point is lower than that in front of the listening point. However, the error $D_{\text{Err}}$ of the proposed method is lower than that of the conventional method in all conditions except for position (h), where it is inferred that the reverberation of the room affects the sound localization. From these results, it can be said that the proposed method can reproduce the 3-D sound image well in

terms of distance and direction compared with the conventional method. Additionally, the error of the distance is still large because the proposed method does not consider the room reverberation [22]. Hence, we will improve the proposed method to overcome this problem.

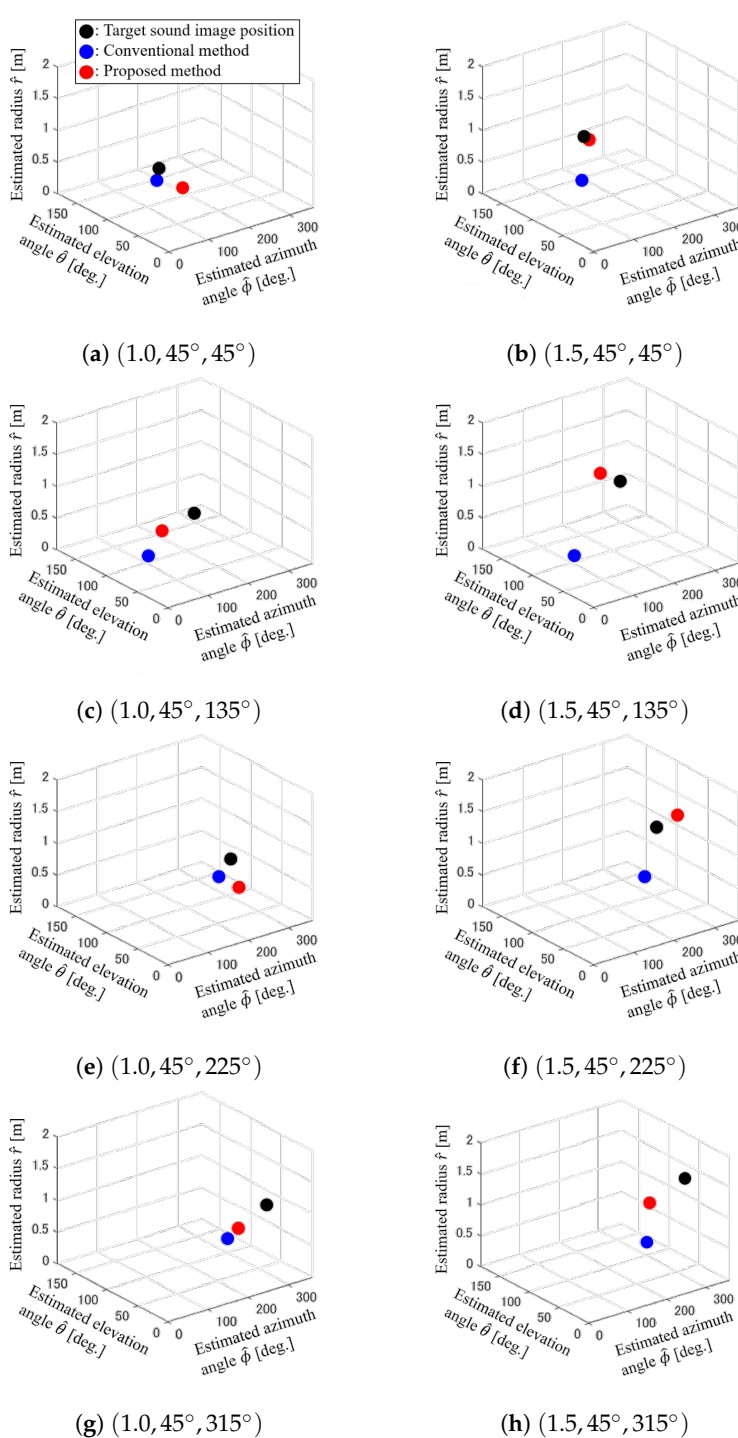

**Figure 11.** Estimated position of 3-D sound image $(\hat{r}, \hat{\theta}, \hat{\phi})$.

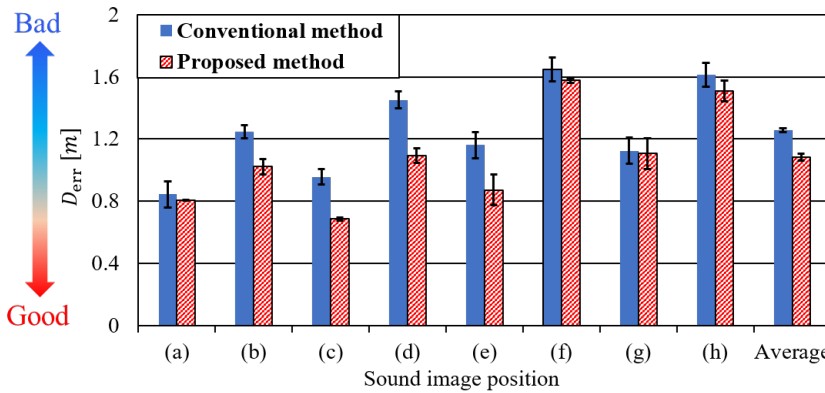

**Figure 12.** Error between the position of the estimated position and that of the target sound image $D_{\mathrm{err}}$.

*4.2. Experiment 2: Clarity of Reconstructed Sound Image for Given Directivity in Proposed Method*

In this experiment, the clarity of the reconstructed sound image for the given directivity in the proposed method was evaluated. The directivities were generated by four different methods; (i) giving only the unit pulse at the target position, (ii) MUSIC, (iii) minimum variance (MV) method [23], and (iv) delay-and-sum (DS) method [24]. These directivities are shown in Figure 13. In this experiment, the directivity patterns were obtained for each position shown in Table 4. Hence, the 32 directivity patterns were used in the experiment. Then, the weight $w_i(k)$ was calculated for each directivity pattern. After that, the 3-D sound image was reconstructed by 22.2 multichannel audio and the emitted sound was recorded by the dummy head microphone. Using the recorded sound obtained at the left and right ears of the dummy head microphone, the inter-aural cross correlation (IACC) [25] was calculated as the evaluation metric. The IACC is obtained by:

$$\mathrm{IACC} = \max_{-1 \le \tau \le 1} \mathrm{IACF}(\tau), \tag{31}$$

$$\mathrm{IACF}(\tau) = \frac{\sum_{t=0}^{t_{\max}-1} s_{\mathrm{R}}(t) s_{\mathrm{L}}(t+\tau)}{\sqrt{\sum_{t=0}^{t_{\max}-1} s_{\mathrm{R}}^2(t) \sum_{t=0}^{t_{\max}-1} s_{\mathrm{L}}^2(t)}}, \tag{32}$$

where $s_{\mathrm{L}}(t)$, $s_{\mathrm{R}}(t)$ are the recorded sounds at the left and right ears, $t_{\max}$ is the length of the signal, $\tau$ is the time index related to the ITD, and $\mathrm{IACF}(\tau)$ is the inter-aural cross function (IACF) for $\tau$. The larger the IACC, the higher the clarity of the reconstructed sound image. In this experiment, we conducted the three trials and the IACC was calculated as the average of three.

Figure 14 shows the IACCs for each position (a)–(h). In Figure 14, "Unit pulse" represents the IACC for the sound image by using the unit pulse at the target position, "MUSIC," "MV," and "DS" represent the IACC for the sound images by using MUSIC, the MV method, and DS, respectively. From Figure 14, the IACCs for Unit pulse, MUSIC, MV, and DS are 0.33, 0.37, 0.36, and 0.35, respectively. This result indicates that the directivity with not only the peak but also the sidelobe affects the high clarity of the sound image in the proposed method. In addition, the clarities of the sound image in front of the listening point are higher than those behind the listening point. This is because the many loudspeakers are placed at the front of the 22.2 multichannel audio and it is easy to reproduce the clear sound image to the front side of the 22.2 multichannel audio. From these results, it can be said that the proposed method is effective for reproducing the 3-D sound image in the 22.2 multichannel audio by using the given directivity pattern with the sidelobe.

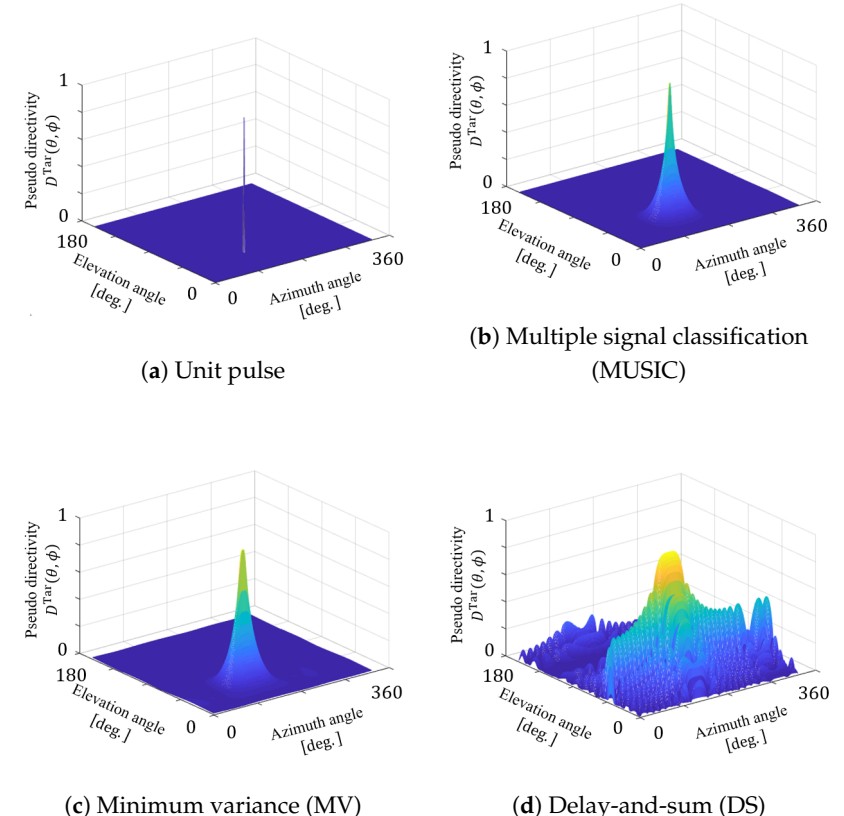

(**a**) Unit pulse

(**b**) Multiple signal classification (MUSIC)

(**c**) Minimum variance (MV)

(**d**) Delay-and-sum (DS)

**Figure 13.** Generated directivities.

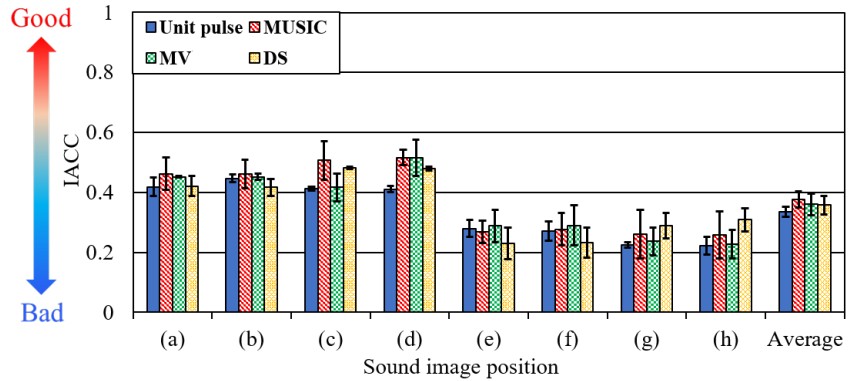

**Figure 14.** Inter-aural cross correlation (IACC) for each generated directivity.

## 5. Conclusions

In this paper, we proposed a novel 3-D sound image reconstruction method for 22.2 multichannel audio based on SH expansion. The proposed method can consider the directivity of the sound image and calculate the filter coefficients by using the mode strength. Hence, it can consider not only the direction but also the distance of the sound image. The experimental results showed that the proposed method can reproduce the 3-D sound image closer to the target position compared with VBAP in terms of direction and distance.

In future, we will develop the proposed method for sounds with higher frequency components above 8000 Hz. This is because the order of the SH expansion depends on the highest frequency of the sound, and the higher the order, the more difficult the SH expansion of the directivity. According to [22], it is effective to consider the reverberation

of the room for reproducing a sound image. Hence, we will consider the reverberation of a room in the proposed method to improve the accuracy of sound localization.Then, we will develop the proposed method to reproduce the 3-D sound images located both inside and outside the 22.2 multichannel audio simultaneously. Thereafter, we will investigate the effectiveness of the proposed method for moving sound images.

**Author Contributions:** T.N. and H.S. conceived the proposed method. H.S. developed the method and conducted the experiments. K.I. wrote this manuscript and modified the figures and expressions of the equations. All authors discussed the results and contributed to the final manuscript. All authors have read and agreed to the published version of the manuscript.

**Funding:** This work was partly supported by JSPS KAKENHI, grant number 19H04142, 21H03488, and 21K18372, and the Ritsumeikan Global Innovation Research Organization (R-GIRO).

**Institutional Review Board Statement:** Not applicable.

**Informed Consent Statement:** Not applicable.

**Data Availability Statement:** Not applicable.

**Conflicts of Interest:** The authors declare no conflict of interest.

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
