# Peer review of "3-D Sound Image Reproduction Method Based on Spherical Harmonic Expansion for 22.2 Multichannel Audio"

_applsci, doi:10.3390/app12041994_

Round 1

Reviewer 1 Report

The article by Kenta Iwai, Hiromu Suzuki and Takanobu Nishiura is devoted to the method proposed by the authors for reproducing a three-dimensional (3-D) sound image, based on the decomposition of a three-dimensional sound image into spherical harmonics. This allows not only to synthesize the direction of an imaginary sound source in a 22.2 multichannel sound system, but also to show an imitation of its distance, using the novel 3-D sound image reproduction method proposed by the authors that controls both the directivity and distance of the target sound image on the basis of SH expansion and mode matching. Such problems are widely known in optical holography. They are based on the synthesis of the hogel-based holographic recording system. (David Blinder Ayyoub Ahar Stijn Bettens Tobias Birnbaum Athanasia Symeonidou Heidi Ottevaere Colas Schretter Peter Schelkens Signal processing challenges for digital holographic video display systems /Signal Processing: Image Communication Volume 70, February 2019, Pages 114-130). It should be noted that today in holography the emphasis is on obtaining continuous parallax. (For example: https://doi.org/10.1364/OE.16.021415) In this regard, I would like to understand how the 22.2 sound system chosen by the authors corresponds to such a criterion as continuous parallax, because this is the characteristic that allows equalizing true light sources, sound, any radiation with a discrete number of synthesized speakers. It would be interesting to hear the views of the authors on this. As for the rest, the calculations carried out by the authors, which are confirmed by experiments, are beyond doubt. The practical usefulness of the work is obvious. I believe that this work fully complies with the requirements of Apple Science magazine.

Author Response

Thank you for taking the time for review of our manuscript and giving the comments.

Kenta Iwai, Hiromu Suzuki, and Takanobu Nishiura

Reviewer 2 Report

The subject of the work is a method of reproducing a three-dimensional sound image to be used in the 22.2 multichannel audio system. Such a sound reproduction system is to be used in the UHDTV standard and, together with a very high definition image, is to faithfully reproduce sound effects. While it is possible to record such multi-channel sound inherently using an array of microphones, methods are also used to synthesize the three-dimensional soundstage by computationally generating a sound wave to be emitted by each of 22 + 2 loudspeakers.
The elementary task in such a synthesis is to obtain the directivity effect, i.e. the impression of sound coming from a specific place in space. Usually, the VBAP (Vector Base Amplitude Panning) method is used here, which in short consists in selecting three speakers that surround a given direction and sending an appropriately weighted sound to these speakers. However, this method only allows to simulate the direction from which the sound is coming. It is not possible to simulate the distance between the virtual sound source and the listener. The authors propose a more complex method based on spherical harmonics (SH), which allows to place the simulated sound source at a selected point in space, taking into account both direction and distance.
Getting the sound distribution to be transferred to a specific point in space for the components to be sent to the individual speakers is not trivial. The calculations are similar to the decomposition of a function into harmonics by the Fourier transform, but here the components are spherical harmonics. The result of the calculation are filter parameters to be applied to the source sound sent to each of the speakers. Thus, it is necessary to drive not three, but all of the speakers, and not using simple signal amplitude scaling, but frequency domain filtering.
The authors carried out tests showing that the effect of locating the apparent source at the indicated point is actually obtained. It is not a perfect representation, but even a directivity comparison shows an advantage over conventional methods.
The article is written clearly.
If the Authors have such data, I would suggest adding a few sentences about the comparison of the computational effort needed in this method with the conventional ones. 

Author Response

Thank you for reviewing our manuscript. Also, thank you for giving us the comment about the computational complexity. Our reply has been attached in the system and please read it. 

Thank you. 

Kenta Iwai, Hiromu Suzuki, and Takanobu Nishiura

Reviewer 3 Report

The paper presents a new interesting approach to the high quality sound reproduction but the quality of the presented materials does not allow to recommend it for publication “as is”.

  1. The presented experimental data do not show the that the suggested methos is superior other the earlier developed VBAP. The both methods demonstrated the large distance errors that are practically same as the distance of target sound image. Any physical experiment where errors are the same as the measured parameters is considered as unsatisfied.
  2. The suggested method ignores the room reverberation even the measured reverberation time is average for control rooms. Reverberation is important to human perception and there are many publications investigating reverberation influence to sound localization. One of the examples is Hz Zheng, K., Otsuka, M. and Nishiura, T., 2019. 3-D sound image localization in reproduction of 22.2 multichannel audio based on room impulse response generation with vector composition. Universitätsbibliothek der RWTH Aachen.
  3. The paper has many negligences. Loudspeaker arrangement (Table 3) does not include distances. The shown frequency band of the band-limited white noise (0–8000) is unrealistic since there are no sources with so low frequency band.
  4. I recommend to include to the paper references about books in binaural and multichannel audio with brief review of science of this direction. Examples of books are: Roginska, A. and Geluso, P. eds., 2017. Immersive sound: The art and science of binaural and multi-channel audio. Taylor & Francis. Geluso, P., 2017. Immersive Sound: The Art and Science of Binaural and Multi-Channel Audio. Taylor & Francis.
  5. Information about practical applications of 22.2 sound system is desirable.
  6. Modern state of art in this area can be presented and the advantages developed method can be compared with the previous methods.

Author Response

Thank you for reviewing our manuscript and giving us the fruitful comments. Our reply has been attached in the system and please read it. 

Thank you. 

Kenta Iwai, Hiromu Suzuki, and Takanobu Nishiura

Round 2

Reviewer 3 Report

The authors improved the paper and it can be published.